# Dietary and Lifestyle Interventions to Mitigate Oxidative Stress in Male and Female Fertility: Practical Insights for Infertility Management—A Narrative Review

**DOI:** 10.3390/metabo15060379

**Published:** 2025-06-08

**Authors:** Efthalia Moustakli, Athanasios Zikopoulos, Periklis Katopodis, Stefanos Dafopoulos, Vasilis Sebastian Paraschos, Athanasios Zachariou, Konstantinos Dafopoulos

**Affiliations:** 1Laboratory of Medical Genetics, Faculty of Medicine, School of Health Sciences, University of Ioannina, 45110 Ioannina, Greece; katopodisper@gmail.com; 2Third Department of Obstetrics and Gynecology, University General Hospital “ATTIKON”, Medical School, National and Kapodistrian University of Athens, 12462 Athens, Greece; thanzik92@gmail.com; 3Department of Health Sciences, European University Cyprus, Nicosia 2404, Cyprus; stefanosntf2001@gmail.com; 4Corewell Health Hospital, Department of Obstetrics and Gynecology, 4700 Schaefer Road Suite 310, Dearborn, MI 48126, USA; vasilispar10@gmail.com; 5Department of Urology, School of Medicine, University of Ioannina, 45110 Ioannina, Greece; zahariou@otenet.gr; 6IVF Unit, Department of Obstetrics and Gynecology, Faculty of Medicine, School of Health Sciences, University of Thessaly, 41110 Larissa, Greece

**Keywords:** oxidative stress, fertility, antioxidants, lifestyle interventions, sperm quality, oocyte health, DNA fragmentation, reproductive biomarkers

## Abstract

**Background/Objectives:** Infertility in both men and women can be significantly influenced by oxidative stress (OS), which occurs due to an imbalance between reactive oxygen species (ROS) and the body’s antioxidant defenses. In women, OS disrupts oocyte maturation, implantation, and the viability of the embryo; in men, it impairs sperm quality, reduces motility, and damages DNA integrity. This review explores existing research on how dietary and lifestyle interventions can reduce OS and enhance reproductive health outcomes. **Methods:** We conducted a comprehensive review of clinical, translational, and molecular studies exploring the mechanisms by which OS affects fertility, as well as the efficacy of nutritional and behavioral strategies. The interventions evaluated include weight management, regular exercise, micronutrient supplementation, antioxidant-rich diets, smoking and alcohol cessation, and stress-reduction techniques. **Results:** Specific dietary components such as zinc, selenium, vitamins C and E, and polyphenols have been found to neutralize reactive oxygen species (ROS) and enhance gamete function. OS is additionally reduced through lifestyle modifications, including minimizing harmful exposures, managing stress, and participating in moderate physical activity. Biomarkers such as ROS levels, total antioxidant capacity, 8-OHdG, and DNA fragmentation index are essential for assessing the effectiveness of interventions. **Conclusions:** Fertility in both sexes can be improved, and oxidative stress significantly reduced, through a multimodal approach incorporating dietary and lifestyle changes. There are encouraging opportunities to improve reproductive health through customized approaches that are informed by biomarker profiles. To incorporate these treatments into regular fertility care, future studies should concentrate on standardized procedures and long-term results.

## 1. Introduction

Oxidative stress (OS) occurs when there is an imbalance between the production of reactive oxygen species (ROS) and the body’s ability to neutralize or repair the harmful effects of these species. While ROS are essential for many physiological functions, excessive concentrations can cause oxidative damage and disrupt cellular equilibrium. OS has been linked to several medical conditions, including male and female reproductive issues [1].

The reproductive system in males is particularly vulnerable to OS due to its unique composition and physiological processes. Spermatozoa are particularly vulnerable to oxidative damage because they have few antioxidant-defense mechanisms. Maintaining normal sperm function and fertility necessitates a delicate balance of ROS production and antioxidant defense. Understanding the influence of OS on male reproductive function is crucial given the rising global frequency of male infertility, with male factors accounting for about 50% of all infertility cases. OS is a potential underlying factor affecting sperm quality, motility, and DNA integrity. Investigating this connection provides valuable insights into the causes and management of male infertility, as well as potential therapeutic targets and strategies to protect and enhance sperm quality [2].

Female fertility, on the other hand, is influenced by a complex interplay of physiological, environmental, and behavioral factors. Infertility is often caused by conditions such as endometriosis, ovulation disorders, chronic diseases, and age-related factors. OS is essential in these disorders because it influences oocyte quality, the follicular environment, and the implantation process. Diet, physical exercise, and environmental pollutants all contribute to increased oxidative damage. Moderate physical exercise and a well-balanced diet high in antioxidants have been demonstrated to enhance ovarian function and overall fertility, especially in women who are obese or have high stress levels [3,4].

Given the significant impact of OS on reproductive health in both genders, a comprehensive approach that includes dietary and lifestyle modifications is essential [5]. This review aims to thoroughly examine the mechanisms by which OS impacts male and female reproductive function. By reviewing the current scientific evidence, we aim to highlight the potential benefits of dietary and lifestyle interventions in reducing OS and enhancing reproductive outcomes. Furthermore, this study will provide doctors and researchers with practical insights to develop targeted strategies for infertility management through antioxidant supplementation and other modifiable factors.

## 2. Methodology

This narrative review aims to provide an overview of what is currently known about how OS contributes to infertility. Through focused searches of electronic databases like PubMed and Google Scholar, relevant literature was found by utilizing keywords like “oxidative stress”, “infertility”, “reactive oxygen species”, “antioxidants”, and “fertility”. Neither rigorous inclusion nor exclusion criteria nor a methodical process governed the search. Articles were selected based on their quality, relevance, and contribution to the field, including original research papers, review articles, and English-language clinical guidelines. This approach allowed a broad overview of the subject while emphasizing key findings and clinical implications.

## 3. Mechanisms Linking OS to Infertility

### 3.1. Roles of OS in Spermatogenesis and Oocyte Quality

To maintain normal sperm function and fertility, a precise balance must be struck between ROS production and antioxidant defense. Understanding the impact of OS on male reproductive function is crucial, especially considering the increasing global incidence of male infertility, which accounts for approximately half of all infertility cases [2].

OS is an important physiological regulator and possible disruptor of reproductive processes, impacting spermatogenesis and oocyte quality. The impact of ROS is determined by the balance of their generation and the ability of antioxidant systems to neutralize them [6].

However, high ROS levels disturb spermatogenesis and jeopardize sperm quality. High ROS concentrations can impair genetic integrity, leading to sperm DNA damage, including fragmentation and chromatin abnormalities, which hinder successful reproduction. Furthermore, ROS can induce lipid peroxidation by oxidizing polyunsaturated fatty acids in sperm membranes, which reduces sperm motility and viability. Excessive OS contributes to male infertility and reduced sperm counts, and it also leads to the death of developing spermatogenic cells. Male reproductive capacity is further compromised by OS, which is often exacerbated by conditions such as varicocele, infections, exposure to environmental pollutants, and poor lifestyle choices [7].

Likewise, OS affects oocyte quality in a significant yet potentially detrimental way. ROS play a role in ovulation, oocyte maturation, and folliculogenesis at regulated levels. For instance, ROS promote the resumption of meiosis in oocytes and aid in the follicular wall’s disintegration during ovulation. These functions demonstrate their physiological importance in the reproduction of females [8].

However, oocytes with high OS can cause serious reproductive issues. It accelerates oocyte aging by damaging mitochondrial DNA, reducing their potential for development and energy production [9]. Elevated ROS levels increase the risk of aneuploidy by interacting with the meiotic spindle and inducing aberrant chromosomal segregation. Oxidative damage can decrease the chances of successful implantation and hinder fetal growth. Oocyte maturation and folliculogenesis are often disrupted in disorders like polycystic ovarian syndrome (PCOS), where OS levels are typically elevated. Similarly, inflammatory reproductive conditions such as endometriosis aggravate OS and reduce oocyte quality [8] (Table 1).

ROS inhibit mitochondrial energy production, creating a vicious cycle of increased ROS formation, a major mechanism behind OS. Additionally, OS-induced DNA damage compromises genomic integrity through the formation of oxidative lesions like 8-OHdG. Moreover, OS affects the structure and function of proteins essential for fertilization and embryo development [10].

Several management strategies can be employed to reduce the deleterious effects of OS on spermatogenesis and oocyte quality. Antioxidants, such as coenzyme Q10, vitamin C, and vitamin E, can neutralize excessive ROS and protect reproductive cells. Lifestyle changes, such as reducing exposure to environmental contaminants, quitting smoking and excessive alcohol consumption, and managing stress, can also help reduce OS levels. Medical treatments that target the underlying causes of oxidative damage, such as anti-inflammatory medicines for endometriosis or medical problems like varicocele, can significantly enhance reproductive health [6,7].

### 3.2. Mechanisms of DNA Fragmentation in Sperm and Implications of Embryo Viability

OS is a leading cause of sperm DNA fragmentation, which severely impacts fertility, embryonic development, and reproductive outcomes. ROS are generated in sperm cells both exogenously (via chemical exposure, radiation, smoking, or viruses) and endogenously (through mitochondrial dysfunction). When ROS levels surpass the antioxidant defenses’ capacity to neutralize them, oxidative damage occurs, leading to structural alterations in sperm DNA, such as base oxidation, strand breaking, and crosslinking. Male infertility is strongly associated with 8-OHdG, a marker of oxidative DNA damage [11].

The unique structure of sperm increases their vulnerability to oxidative DNA fragmentation. During spermiogenesis, protamines replace histones, tightly condensing the chromatin, which protects it from physical damage but limits the sperm’s ability to repair DNA. Unlike somatic cells, sperm lack the enzymatic tools to recognize and fix oxidative lesions, making them more susceptible to accumulated oxidative damage. This compromises the sperm’s motility, its ability to fertilize an egg, and the genetic material it passes on to the oocyte [12,13].

Sperm DNA fragmentation after conception impacts embryonic development. During early cell divisions, the zygote heavily relies on paternal DNA, and excessive fragmentation can lead to chromosomal instability, developmental arrest, or failure to reach the blastocyst stage. Furthermore, embryos derived from sperm with damaged DNA have a diminished capacity to repair oxidative damage in the early stages of development, which may result in long-term genetic abnormalities and increase the risk of congenital disorders, miscarriage, or failed implantation [14].

Sperm DNA fragmentation further complicates ART. Although intracytoplasmic sperm injection (ICSI) and in vitro fertilization (IVF) address issues related to sperm morphology and motility, they cannot mitigate the effects of DNA fragmentation. In ART cycles, higher levels of sperm DNA fragmentation have been linked to lower live birth rates, decreased fertilization rates, and inferior embryo quality. ICSI with sperm containing fragmented DNA has been associated with a higher risk of pregnancy loss and inadequate newborn outcomes [15].

To improve male reproductive health, targeted therapy that reduces ROS levels is needed. Antioxidant therapy, including vitamins C and E, coenzyme Q10, zinc, and selenium, effectively reduces oxidative DNA damage. Lifestyle changes, such as stress management, reducing alcohol consumption, and quitting smoking, can also help lower OS and improve sperm quality. Addressing primary concerns such as varicocele, infections, or obesity can help to minimize OS and improve reproductive results [16].

### 3.3. Mitochondrial Dysfunction as a Common Pathway

Infertility and OS are closely linked, as mitochondria serve as both the cell’s powerhouses and major producers of ROS. Oxidative phosphorylation, the process by which mitochondria generate energy, inevitably produces ROS as a byproduct. Healthy cells possess robust antioxidant systems that neutralize ROS and sustain cellular homeostasis. However, when mitochondrial function is compromised by aging, environmental pollutants, infections, lifestyle choices, or metabolic disorders, the balance between ROS production and antioxidant defense is disrupted. This overproduction of ROS causes OS, which can have serious consequences for reproductive health [13,17].

In males, mitochondrial dysfunction significantly affects sperm quality. For successful fertilization, mitochondria must generate the ATP needed for sperm motility. When mitochondrial activity is compromised, the ability of sperm to swim toward and penetrate the oocyte is limited. Furthermore, sperm membranes are subjected to oxidative damage as a result of lipid peroxidation caused by excess ROS produced by dysfunctional mitochondria. This further reduces their structural integrity and motility [18]. Furthermore, ROS can cause sperm DNA fragmentation, which damages genetic material and jeopardizes the genetic integrity required for conception and early embryonic development. Due to their poor DNA repair systems, sperm frequently acquire oxidative DNA damage, resulting in infertility or the transmission of genetic abnormalities to progeny [13].

Females are more vulnerable to mitochondrial malfunction, particularly in oocytes. During the intricate processes of maturation, fertilization, and early embryonic development, oocytes primarily depend on mitochondrial function to produce energy. Oocytes are more susceptible to OS and mitochondrial dysfunction due to having fewer mitochondria than other cells and a limited ability to repair mitochondrial damage [19]. Age-related OS can lead to the gradual deterioration of mitochondrial DNA in a woman’s oocytes, reducing ATP synthesis. This decrease in energy availability inhibits essential cellular activities, including meiotic division, spindle formation, and chromosomal segregation, therefore raising the likelihood of aneuploidy and other developmental problems. Additionally, it has been shown that oocyte mitochondrial dysfunction contributes to infertility by diminishing oocyte quality, decreasing fertilization rates, and increasing miscarriage rates [20].

Additionally, mitochondrial dysfunction affects both the developing embryo and individual gametes. In the early stages of embryonic development, the zygote depends on paternal mitochondrial DNA provided by the sperm. However, ROS-induced damage may have already damaged the mitochondrial environment in sperm, resulting in poor embryonic development. Furthermore, mitochondrial failure may impair the embryo’s ability to repair oxidative DNA damage during early cell divisions, increasing genetic instability and the risk of developmental arrest or implantation failure [21].

## 4. Dietary Interventions

Reducing OS through nutritional therapy is critical for boosting fertility in both men and women. OS occurs when the body’s antioxidant defenses are out of sync with the production of ROS. This type of imbalance has the potential to compromise reproductive hormone function, gamete quality, and conception effectiveness. The primary goals of nutritional therapy aimed at lowering OS are to increase antioxidant consumption, promote metabolic health, and address underlying dietary deficiencies [22].

Antioxidant-rich food is crucial for scavenging ROS. Berries, citrus fruits, spinach, kale, nuts, seeds, and other fruits and vegetables rich in beta-carotene, polyphenols, and vitamins C and E are important sources. Foods high in zinc (such as pumpkin seeds and shellfish) and selenium (such as Brazil nuts) promote antioxidant enzymes like superoxide dismutase and glutathione peroxidase. 

Omega-3 fatty acids, which are present in plant sources like flaxseeds and chia seeds and fatty seafood like salmon and mackerel, have anti-inflammatory qualities, support the integrity of cell membranes, and encourage hormonal balance. These advantages are associated with better oocyte health in females and sperm quality in males [23].

Micronutrients like iron, vitamin D, and folate are essential for reproductive health. Iron lowers OS in reproductive tissues, vitamin D helps hormonal balance, and folate facilitates DNA synthesis. In cases where inadequacies exist, supplementation can enhance results.

Additional antioxidant protection is provided by phytochemicals such as carotenoids and flavonoids. Reduced OS and better sperm quality are associated with lycopene and resveratrol. Maintaining glycemic control is critical, particularly in individuals with conditions like PCOS [24]. Diets low in refined carbohydrates and high in complex carbs, fiber, and protein stabilize blood sugar levels and reduce systemic inflammation. Whole grains, legumes, and non-starchy vegetables are beneficial. Avoiding processed foods, trans fats, and excessive alcohol is essential, as these can exacerbate OS. Personalized dietary interventions, guided by biomarkers and nutrigenomics, can optimize antioxidant intake and address individual needs (Table 2, Figure 1). Consequently, dietary strategies incorporating antioxidants, omega-3s, essential micronutrients, and phytochemicals, along with glycemic control and the avoidance of harmful practices, significantly enhance reproductive health in both men and women [25].

## 5. Lifestyle Interventions

Making lifestyle changes is essential for preventing OS, a significant contributor to infertility in both men and women. OS results from an imbalance between the generation of ROS and the body’s antioxidant defense mechanisms. Key lifestyle measures, including regular exercise, weight management, smoking and alcohol cessation, and stress reduction, have been shown to lower OS while improving reproductive health. These therapies not only address the underlying causes of oxidative damage but also enhance hormonal balance, improve gamete quality, and support overall fertility outcomes [26].

### 5.1. Regular Excerise

Participating in frequent, moderate physical activity is an excellent way to reduce OS and increase fertility. Exercise boosts the body’s overall antioxidant defenses, reduces inflammation, and improves metabolic health, all of which are essential for reproductive function. Regular exercise enhances blood circulation, optimizing oxygen and nutrient delivery to reproductive organs and thereby promoting their optimal function and protecting against oxidative damage [27]. Current health and fertility guidelines advise engaging in at least 150 min of moderate-intensity aerobic activity per week, which equates to about 30 min a day, five days a week. Activities such as yoga, brisk walking, swimming, cycling, and light to moderate resistance training are all suitable forms of exercise. These forms of exercise have been shown to enhance reproductive outcomes in both men and women, support hormonal balance, and decrease systemic OS [28].

However, it is critical to maintain a balance, as excessive or strenuous activity might have negative consequences. Prolonged or vigorous physical exercise raises ROS levels, alters hormonal balance, and may decrease ovulatory function and sperm quality. A balanced exercise plan, such as brisk walking, yoga, swimming, or weight training, can help men and women reduce OS and improve reproductive health. A personalized approach, guided by a healthcare professional, ensures that the type and intensity of exercise align with the individual’s needs and fertility goals [29].

### 5.2. Weight Management

Achieving and maintaining a healthy body weight is essential for boosting fertility, since obesity and being underweight are linked to OS and hormonal imbalances. In obese individuals, excess adipose tissue produces elevated levels of ROS and inflammatory cytokines, influencing sperm motility, increasing DNA fragmentation, and disrupting ovarian function. A long-term strategy for weight control is critical for restoring oxidative balance and improving reproductive health [30]. A nutrient-dense, antioxidant-rich diet combined with regular physical activity can help regulate body weight, minimize OS, and improve reproductive success. Even modest weight loss in overweight or obese individuals has been shown to improve sperm quality, restore ovulatory cycles, and enhance the success rates of ART. According to clinical guidelines, the ideal body mass index (BMI) for fertility is between 18.5 to 24.9 kg/m^2^. Moreover, waist circumference is an effective indicator of central fat accumulation and the related risk of metabolic disorders. In reproductive-aged individuals, waist circumference should ideally be below 102 cm for men and below 88 cm for women. These targets improve hormonal balance and reproductive function by lowering inflammation and OS [31]. A steady increase in calorie intake through appropriate food choices can restore hormonal balance and improve reproductive capacity in underweight people [30,32].

### 5.3. Plant-Based Diets and Nutraceutical Support

Plant-based diets focusing on fruits, vegetables, whole grains, legumes, nuts, and seeds are rich in nutraceuticals and functional foods with powerful antioxidant and anti-inflammatory effects [33]. Essential micronutrients such as zinc, selenium, and vitamins C and E, together with phytochemicals like polyphenols, flavonoids, and carotenoids, are present in these diets and play a key role in reducing OS and neutralizing reactive oxygen species. This is especially important when it comes to infertility, as oxidative stress is known to contribute to hormone imbalance and poor gamete quality [34]. Research has indicated that people who follow plant-based or Mediterranean-style diets have better reproductive results. For example, studies have reported that a diet high in antioxidants is positively correlated with fertility parameters, while other research has highlighted the role of plant-derived bioactives in enhancing ovarian function, sperm quality, and hormonal balance [35,36]. These dietary patterns confer a range of clinically relevant benefits, including the reduction of systemic inflammation, enhancement of insulin sensitivity, mitigation of OS in reproductive cells, modulation of endocrine function, and improvement of gut microbiota composition and diversity [37]. Current evidence supports the inclusion of plant-based nutritional strategies as a low-risk, non-pharmacological intervention to complement infertility treatment and improve overall reproductive health, even though more high-quality clinical trials are required to establish causality and define optimal dietary patterns [38].

### 5.4. Smoking and Alcohol Cessation

Tobacco and alcohol use are major contributors to OS and reduced fertility in both men and women. Smoking introduces harmful compounds such as cadmium, nicotine, and polycyclic aromatic hydrocarbons, which elevate ROS levels, impair sperm DNA integrity, and accelerate ovarian aging [39]. Smoking cessation improves sperm motility and morphology, preserves ovarian reserve, and markedly reduces exposure to ROS. Similarly, drinking alcohol can reduce fertility by causing hormonal imbalances, elevated OS, and poor gamete quality. Alcohol consumption, whether acute or chronic, elevates inflammatory markers and impairs the body’s antioxidant defense mechanisms [40].

Regarding fertility, no universally recognized limit establishes a “safe” amount of alcohol consumption. However, consumption exceeding one standard drink per day for women and two for men has been repeatedly related to impaired reproductive outcomes. A typical drink consists of about 14 g of pure alcohol, commonly present in 150 mL of wine (12%), 350 mL of beer (5%), or 45 mL of distilled spirits (40%) [41]. During fertility treatments and attempts to conceive, consuming four or more drinks per occasion for women and five or more for men is strongly discouraged because it increases OS. Reducing or eliminating alcohol consumption is recommended to enhance ART outcomes, promote hormonal balance, and support overall reproductive health [42].

### 5.5. Stress Reduction and Techniques

Chronic stress is a major, though often underestimated, factor contributing to OS and infertility. It activates the hypothalamic–pituitary–adrenal (HPA) axis, resulting in the excessive production of cortisol and other stress hormones. Increased cortisol levels disturb hormonal regulation, hindering critical processes such as ovulation, sperm production, and embryo implantation. Furthermore, stress-induced OS can directly damage reproductive cells and tissues, compounding the challenges faced by individuals with infertility [33].

Chronic inflammation and OS are closely related, and lifestyle choices, such as diet, can significantly impact it them. The Mediterranean diet is well known for its anti-inflammatory and antioxidant properties. It is characterized by moderate consumption of fish and poultry and a high intake of fruits, vegetables, whole grains, legumes, nuts, and olive oil [43]. Its rich content of vitamins, omega-3 fatty acids, and polyphenols supports reproductive function by reducing systemic OS and regulating stress hormone levels. According to research, adherence to the Mediterranean diet is associated with better reproductive outcomes in both men and women and lower levels of OS markers [44,45].

Incorporating stress-reduction strategies into daily life can significantly improve reproductive outcomes. It has been demonstrated that mindfulness-based activities, including yoga, meditation, and deep breathing techniques, increase fertility and decrease OS indicators. Counseling and cognitive behavioral therapy can help individuals develop coping strategies, easing the psychological effects of infertility. Acupuncture, massage, and relaxation techniques are examples of complementary therapies that may improve blood flow to reproductive organs, regulate stress reactions, and support hormonal balance [46].

By addressing both the psychological and physiological effects of stress, these treatments reduce oxidative damage and create a more favorable environment for conception. Incorporating stress management techniques into fertility treatment plans can holistically improve reproductive health and overall well-being (Table 2, Figure 2) [47].

## 6. Gut Microbiome, OS, and Fertility

It is becoming increasingly acknowledged that the gut microbiota plays a crucial role in regulating inflammation, systemic oxidative stress, and hormonal homeostasis—all of which are essential for reproductive health. A balanced and varied microbiome supports the HPG axis, regulates immunological responses, and aids in metabolic regulation. A number of infertility-related disorders, such as endometriosis, unexplained male infertility, and PCOS, have been linked to dysbiosis, or disturbances in this microbial equilibrium [48].

Recent research highlights how dietary patterns and physical activity can significantly influence the composition and function of the gut microbiota [49]. In particular, Lactobacillus and Bifidobacterium thrive in diets high in fiber, polyphenols, and prebiotics, such as those found in plant-based or Mediterranean-style diets. These microbes generate short-chain fatty acids (SCFAs), which possess strong antioxidant and anti-inflammatory properties. SCFAs can enhance endometrial receptivity, reduce oxidative damage to reproductive tissues, and contribute to a more favorable hormonal environment. These bacterial metabolites are essential for reducing OS and promoting healthy reproduction. In addition to being shaped by diet, the gut microbiota may provide a strategic focus for upcoming reproductive therapies [50,51].

Moderate, regular physical activity complements dietary influences by promoting microbial diversity and improving gut barrier integrity. Both excessive physical activity and sedentary behavior can disrupt microbial balance and increase systemic inflammation [52]. Since the gut microbiota plays a central role in connecting nutrition, OS, and reproductive health, these findings highlight the necessity of considering it in lifestyle-based fertility interventions. Targeting the microbiota through personalized diet and exercise plans may represent a novel, non-invasive approach to enhancing reproductive outcomes [53].

## 7. Potential Biomarkers for Measuring Efficacy

Reliable biomarkers indicative of changes in oxidative status and reproductive health are crucial for assessing the efficacy of dietary and lifestyle interventions to reduce OS. These biomarkers are essential for assessing the physiological and clinical effects of therapies, guiding treatment strategies, and monitoring progress in infertile patients. Numerous well-studied biomarkers offer critical insights into antioxidant capacity and oxidative damage in both male and female populations [6,7].

### 7.1. ROS in Seminal Plasma

Assessing the efficacy of dietary and lifestyle treatments to reduce OS requires accurate biomarkers that reflect changes in oxidative status and reproductive health. These biomarkers are crucial for evaluating the clinical and physiological responses to medications, guiding treatment strategies, and monitoring the progression of infertility in patients. Multiple widely recognized markers have been rigorously examined, yielding essential information about oxidative damage and antioxidant capacity in both sexes. ROS levels in semen provide a reliable measure of OS within the male reproductive tract [54].

Male infertility is intimately linked to elevated ROS levels because these substances can harm sperm membranes, proteins, and DNA. This damage results in abnormal sperm morphology, reduced motility, and impaired function, ultimately leading to infertility [6]. Given that ROS levels provide crucial insights into OS in seminal plasma, their measurement is essential for evaluating the effectiveness of treatments designed to reduce ROS production or enhance antioxidant defenses. Increased sperm quality is closely correlated with significant reductions in ROS levels, which can be attained by adopting a diet high in antioxidants or changing one’s lifestyle by quitting smoking (Table 3) [55].

### 7.2. Total Antioxidant Capacity (TAC) in Sperm

Enzymatic and non-enzymatic antioxidants’ combined ability to neutralize ROS is reflected in the serum’s total antioxidant capacity (TAC). As an essential marker of the body’s antioxidant defense, TAC levels in infertile individuals are often lower due to the loss of antioxidants from sustained OS. Further cellular and molecular damage in reproductive tissues results from this loss, which hinders the body’s ability to control ROS. TAC measurement provides a comprehensive evaluation of the body’s oxidative state and is valuable for assessing treatments designed to boost antioxidant levels. Changes in lifestyle, including a nutrient-rich diet and consistent moderate exercise, have been demonstrated to elevate TAC, leading to enhanced reproductive outcomes and improved systemic oxidative balance [56].

### 7.3. 8-Hydroxy-2′-deoxyguanosine (8-OHdG) as a Marker of Oxidative DNA Damage

Oxidative DNA damage is commonly identified by the particular biomarker 8-OHdG. It reflects oxidative modifications to the guanine residues in DNA, which are particularly vulnerable to damage by ROS. The adverse effect of elevated 8-OHdG on cellular viability and function has led to a high correlation with infertility. While elevated levels of 8-OHdG are linked to impaired egg integrity and embryo development in females, they are also connected with decreased sperm quality in males. Measuring 8-OHdG levels provides crucial insights into the extent of oxidative DNA damage and the effectiveness of treatments designed to reduce OS. Reproductive health improves when dietary and lifestyle changes that effectively reduce OS also result in significant decreases in 8-OHdG levels [57,58].

### 7.4. Sperm Mitochondrial Activity

Mitochondria are the primary energy producers in sperm and are also significant sources of ROS production. Mitochondrial dysfunction is closely associated with OS and impaired sperm function, including reduced motility and fertilizing capacity. Assessing mitochondrial activity, often through mitochondrial membrane potential (MMP) assays, provides insights into the energy production capabilities and oxidative balance within sperm cells [59]. Improvements in mitochondrial activity following interventions, such as dietary antioxidant supplementation or weight management, suggest reduced OS and enhanced sperm function. As mitochondrial health is directly linked to sperm viability and fertilization potential, this biomarker serves as an important measure of intervention efficacy in addressing male infertility [60].

### 7.5. Sperm DNA Fragmentation (DFI)

The sperm DNA fragmentation index (DFI) assesses the extent of DNA fragmentation in sperm, a condition frequently linked to OS. Elevated DFI levels have been associated with various reproductive challenges, including recurrent pregnancy loss, poor embryo quality, implantation failure, and reduced fertilization rates [61]. DFI serves as a supplementary tool for evaluating sperm quality, alongside conventional indicators like motility and morphology. Strategies that reduce OS, including lifestyle adjustments, antioxidant use, and stress management, frequently lead to better DFI outcomes. DFI levels serve as a key biomarker for evaluating the efficacy of male infertility treatments, as they are frequently associated with improved pregnancy rates, enhanced embryo quality, and greater fertilization success (Table 4) [62].

## 8. Clinical and Translational Insights

The integration of dietary and lifestyle recommendations into fertility treatment protocols offers a promising avenue for enhancing treatment efficacy, improving patient outcomes, and addressing the underlying causes of OS. While advances have been made in understanding the role of OS in infertility, practical implementation of these insights into clinical practice remains an evolving field [63,64].

### 8.1. Incorporating Dietary and Lifestyle Recommendations into Fertility Treatments

Research shows that implementing targeted dietary and lifestyle changes can significantly enhance the success of ART, including ICSI and IVF. By including antioxidants such as vitamins C and E, coenzyme Q10, and omega-3 fatty acids in fertility treatments, oxidative damage to gametes can be reduced, potentially boosting ART success rates. Additional benefits may arise from personalized antioxidant regimens tailored to an individual’s specific OS biomarkers [65].

Structured lifestyle counseling that incorporates stress reduction, weight management, limiting alcohol intake, and quitting smoking as part of preconception care has been shown to optimize the reproductive window and improve the likelihood of natural conception [66]. A more integrated and effective model of care can be achieved when fertility doctors partner with dietitians, physical therapists, and mental health specialists. Moreover, implementing a pre-treatment optimization phase that includes nutritional and lifestyle modifications before initiating ART may improve gamete quality and implantation success. For example, pre-treatment weight loss in individuals with obesity has been associated with improved hormonal balance, reduced OS, and higher clinical pregnancy rates (Figure 3 and Figure 4) [67].

### 8.2. Translational Challenges and Gaps

To effectively incorporate food and lifestyle modifications into reproductive therapy, several obstacles and gaps must be filled, notwithstanding promising data [68]. The lack of generally recognized guidelines for implementing lifestyle modifications in reproductive care is a serious obstacle, although the advantages of such modifications are well established [69]. Developing evidence-based guidelines that are tailored to diverse patient populations is a critical research priority. Furthermore, although biomarkers like DNA DFI, ROS, and TAC provide insightful information, their routine application in clinical practice is constrained by a lack of standardization, expense, and the requirement for technical competence. Creating assays that are both broadly available and reasonably priced is crucial to efficiently tracking the effects of interventions [70].

Patient adherence poses another significant challenge; sustaining long-term lifestyle changes can be difficult without proper support. Behavioral strategies, including the use of digital health tools, educational initiatives, and ongoing consultations, may help improve compliance and outcomes [71]. The importance of a tailored approach is emphasized by the differences in OS levels, genetic factors, and environmental exposures between individuals. Additional research is needed to identify which patient subgroups would benefit most from targeted interventions. Despite the promising short-term improvements in reproductive outcomes, the lack of long-term data on the effects of these therapies on offspring health and future pregnancies remains a key concern. The successful incorporation of food and lifestyle modifications into standard reproductive treatment depends on filling these gaps (Table 5) [72].

### 8.3. Future Directions

Future research should focus on randomized controlled trials evaluating the effectiveness of combined dietary and lifestyle therapies for various infertility subtypes, including male factor infertility, PCOS, and unexplained infertility, to address these gaps [73]. The collaboration of public health professionals, researchers, and doctors can help facilitate the development of standardized, patient-centered protocols. Additionally, leveraging breakthroughs in precision medicine, such as genomic and metabolomic analysis, may enable the creation of tailored therapies to optimize fertility outcomes [73].

By addressing these challenges and incorporating dietary and lifestyle changes into fertility treatments, a comprehensive and effective approach to managing infertility and improving overall reproductive health can be achieved [64,70].

## 9. Limitations of the Study

As a narrative review, this study does not employ a systematic search or selection process, which may introduce selection bias and affect reproducibility. The literature included reflects the authors’ choices and may not encompass all relevant or recent studies, potentially limiting comprehensiveness. The reviewed studies vary widely in design, population, and outcomes, restricting the ability to directly compare results or perform quantitative synthesis. Additionally, this review does not include a formal quality assessment or risk of bias evaluation of the included articles, which may influence the reliability of conclusions. Given the rapidly evolving nature of research on oxidative stress and fertility, some emerging findings may not be fully covered.

## 10. Conclusions

OS plays a significant role in impairing male and female fertility, with growing evidence highlighting its detrimental effects on gamete quality, embryo development, and overall reproductive health [74]. According to recent studies, the NLRP3 inflammasome is a key modulator of inflammatory reactions that are intimately associated with oxidative stress. In particular, OS and NLRP3 inflammasome activation interact extensively in the ovarian microenvironment, and their combined effects have a considerable impact on reproductive success. This inflammasome represents a promising target for innovative bioactive compounds aimed at mitigating inflammation-induced reproductive damage [75].

Dietary and lifestyle therapies, such as antioxidant-rich meals, regular physical activity, stress management, and avoidance of adverse environmental exposures, have emerged as viable techniques to reduce oxidative damage and enhance reproductive outcomes. Nutrients such as vitamins C and E, zinc, selenium, and polyphenols have potent antioxidant effects, and lifestyle changes, including maintaining a healthy weight, minimizing alcohol and tobacco intake, and implementing stress-reduction techniques, contribute to overall reproductive well-being [6].

Despite the encouraging results, further well-designed clinical trials are required to determine appropriate food and lifestyle guidelines for different populations. Future research should focus on personalized techniques that account for genetic predispositions, gender-specific responses, and the possible synergistic effects of several medications. Clinicians and individuals can improve male and female fertility and reproductive health by implementing evidence-based nutritional and lifestyle changes [76].

## Figures and Tables

**Figure 1 metabolites-15-00379-f001:**
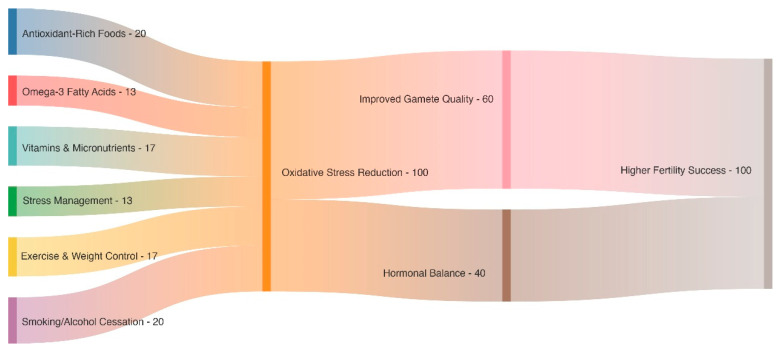
This Sankey diagram illustrates how a number of beneficial therapies, including exercise, stress management, omega-3 fatty acids, and diets high in antioxidants, combine to lessen oxidative stress. Hormonal balance and gamete quality are both improved by this lowering, which eventually increases fertility results.

**Figure 2 metabolites-15-00379-f002:**
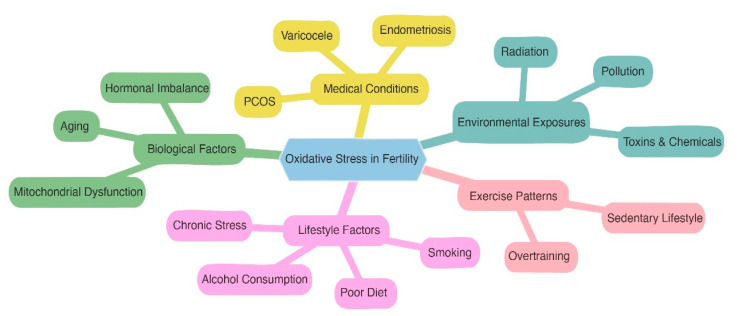
The main causes of oxidative stress in fertility are arranged in the mindmap as follows: poor lifestyle behaviors, biological issues, medical conditions, environmental exposures, and exercise habits. It displays the intricate network of influences.

**Figure 3 metabolites-15-00379-f003:**
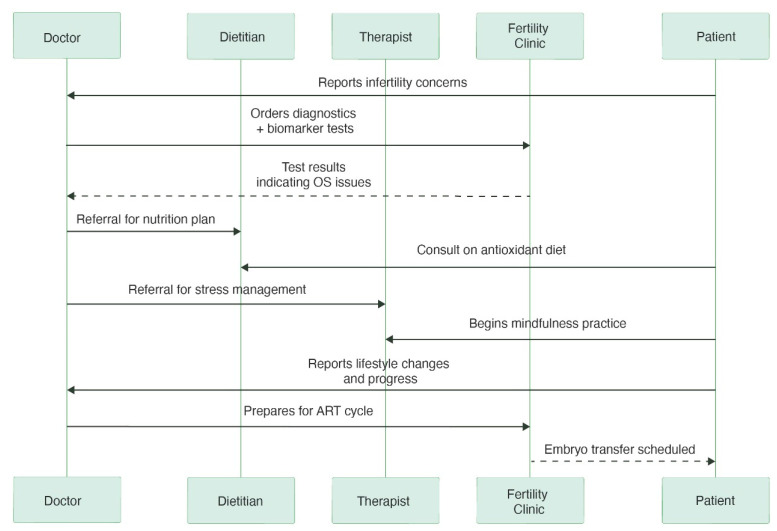
The sequence diagram depicts the patient’s journey toward reproductive treatment. It charts interactions between the patient, physician, dietician, therapist, and fertility clinic, highlighting the ways in which medical interventions, stress management, and lifestyle changes all contribute to being ready for ART. To gain clinically meaningful insights into the causes of infertility, the doctor may prescribe specific biomarkers as part of the diagnostic phase, such as metabolic indicators, hormone tests, or oxidative stress markers (e.g., 8-OHdG, MDA). Some biomarkers are promising, but their accessibility and cost-effectiveness are still being assessed and may differ depending on the context.

**Figure 4 metabolites-15-00379-f004:**
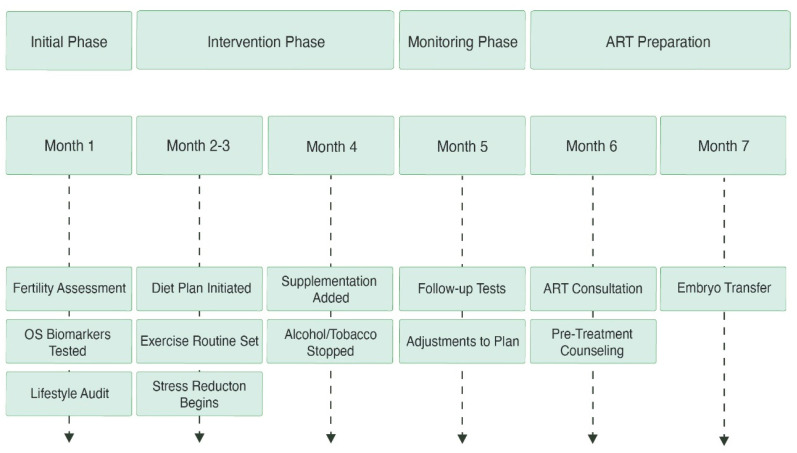
This timeline outlines the stages of fertility optimization over several months. Starting with lifestyle audits and diagnostic testing, it moves through phases of food, exercise, and supplements, includes monitoring and modifications, and ends with getting ready for embryo transfer through ART procedures.

**Table 1 metabolites-15-00379-t001:** This table summarizes the main ROS involved in infertility pathogenesis, detailing their specific roles and how they affect the quality of oocytes and sperm.

ROS Type	Role in Infertility Pathogenesis	Effect on Oocyte Quality	Effect on Sperm Quality
Superoxide anion (O_2_•−)	Primary ROS produced in mitochondria; initiates oxidative damage	Causes mitochondrial dysfunction, DNA fragmentation, and apoptosis	Induces sperm DNA fragmentation and reduces motility
Hydrogen peroxide (H_2_O_2_)	Stable ROS that diffuses through membranes, propagates OS	Impairs oocyte maturation and reduces fertilization capacity	Decreases sperm motility and damages membrane integrity
Hydroxyl radical (•OH)	Highly reactive; causes lipid peroxidation and DNA damage	Damages oocyte membrane lipids, affecting viability	Causes lipid peroxidation in sperm membranes, reducing fluidity
Nitric oxide (NO)	Modulates vascular tone and sperm capacitation	In excess, disrupts meiotic progression and induces apoptosis	Alters sperm motility and capacitation at high concentrations
Peroxynitrite (ONOO−)	Reactive nitrogen species formed from NO and superoxide	Promotes oxidative DNA damage and mitochondrial impairment	Induces oxidative damage leading to decreased sperm function

**Table 2 metabolites-15-00379-t002:** The main dietary components and their main sources are listed in this table, with a focus on how they enhance fertility by lowering oxidative stress and enhancing reproductive health. Hormonal balance, gamete quality, and overall reproductive outcomes can all be improved by eating a healthy, nutrient-rich diet and abstaining from dangerous substances.

Nutritional Components	Main Sources	Benefits in Fertility
Antioxidants	Fruits (berries, citrus), Vegetables (spinach, kale) Nuts (almonds, walnuts)	Neutralize ROS, protect reproductive tissues
Selenium and zinc	Brazil nuts Pumpkin seeds Shellfish	Support enzymatic antioxidants such as glutathione peroxidase
Omega-3 fatty acids	Fatty fish (salmon, mackerel) Flaxseeds Chia seeds	Improve sperm and oocyte quality, reduce inflammation
Micronutrients	Folate (leafy greens, fortified cereals) Vitamin D (sun exposure, fatty fish) Iron (lean meats)	Enhance hormonal regulation (vitamin D, iodine), DNA synthesis and repair (folate, B12), oxygen transport (iron), and antioxidant enzyme function (magnesium) Essential for oocyte and sperm development; deficiencies linked to anovulation, poor sperm quality, and implantation failure
Phytochemicals	Resveratrol (red grapes, berries) Lycopene (tomatoes)	Reduce OS, improve sperm quality
Low-glycemic diet	Whole grains Legumes Non-starchy vegetables	Stabilizes blood glucose and insulin levels, reduces systemic inflammation and oxidative stress; particularly beneficial for women with PCOS and men with metabolic syndrome; enhances hormonal balance, ovulation, and sperm function
Avoidance	Processed foods, Trans fats Excessive alcohol	Reduces ROS production, maintains antioxidant capacity

**Table 3 metabolites-15-00379-t003:** Through a variety of biological processes, the lifestyle modifications highlighted in this table reduce oxidative stress and increase fertility. Evidence supporting moderation, sustainability, and behavioral change supports the quantifiable effects of each strategy, from stress reduction to exercise, on reproductive health.

Intervention	Action Mechanism	Impact on Fertility	Evidence and Notes
Regular exercise	Enhances circulation and strengthens antioxidant defense	Improves ovulation, hormonal balance, and sperm quality	Prevent ROS by avoiding excessive exercise
Weight management	Reduces OS and hormonal imbalances	Enhances ovulation, ART success, and sperm parameters	Sustainable approaches are essential
Plant-based diet and Nutraceuticals	Provides antioxidants and anti-inflammatory compounds; reduces systemic OS	Improves sperm quality, ovarian function, and hormonal balance	Rich in polyphenols, flavonoids, and vitamins C/E; supported by recent clinical studies
Smoking cessation	Reduces ROS exposure and DNA damage	Decreases the depletion of ovarian reserves and increases sperm motility	Post-cessation benefits are substantial
Alcohol reduction	Lowers systemic inflammation and prevents the depletion of antioxidants	Increases implantation rates and the quality of sperm and oocytes	Limit even moderate alcohol consumption
Stress reduction	Balances the HPG axis and reduces cortisol	Improves the integrity of sperm DNA, ovulation cycles, and implantation	Includes yoga, mindfulness, and other beneficial techniques

**Table 4 metabolites-15-00379-t004:** The main oxidative-stress-related indicators used in reproductive medicine are listed in this table along with information on how to measure them, their clinical importance, and evaluation methods. Providing important information on gamete quality and fertility outcomes, these indicators aid in the assessment of oxidative damage and antioxidant capability.

Biomarkers	Measurements	Clinical Relevance	Assessment Method
ROS in seminal plasma	Levels of ROS	The male reproductive tissues exhibit signs of OS	Chemiluminescence assays
TAC in serum	Total antioxidant capacity	Represents the antioxidant defense system	Spectrophotometry
8-OHdG	Oxidative DNA damage	Predicts the health of sperm and oocytes by identifying DNA damage in gametes	ELISA, HPLC, or immunoassays
Sperm mitochondrial activity	Mitochondrial function in sperm	Indicates spermatozoa’s OS and energy metabolism	Mitochondrial membrane potential (MMP) assay
Sperm DFI	The proportion of sperm DNA fragments	Associated with miscarriage, embryo quality, and fertilization success	TUNEL assay or sperm chromatin dispersion (SCD)

**Table 5 metabolites-15-00379-t005:** This table summarizes key aspects of oxidative stress (OS) management in fertility care, highlighting their benefits, current challenges, and potential solutions. It underscores the importance of personalized, biomarker-driven approaches and comprehensive lifestyle interventions to improve reproductive outcomes. N/A: not applicable.

Aspect	Crucial insight	Challenges	Proposed Solutions	Typical Dosage
Antioxidant supplementation	Decreases oxidative damage and improves gamete quality	Absence of customized regimens	Personalize supplements using biomarkers	Vitamin C: 500–1000 mg/day Vitamin E: 200–400 IU/day Coenzyme Q10: 100–300 mg/day L-carnitine: 1–3 g/day Zinc: 15–30 mg/day
Lifestyle counseling	Enhances patient adherence to stress, alcohol, and smoking cessation	Absent assistance, adherence may be limited	Include follow-up sessions and digital tools	N/A
Pre-treatment enhancement	Weight loss reduces OS and promotes hormonal balance	Pre-treatment counseling in not always offered	Consider prenatal care a regular occurrence	N/A
Biomarker application	Enables the measurement of OS levels in real time	High cost and limited accessibility	Develop cost-effective, standardized assays	N/A
Personalized interventions	Adapts care to each patient’s needs	Requires combining metabolic and genetic data	Invest in research in precision medicine	Dosages tailored per patient based on metabolic/genetic profile

## Data Availability

Data are unavailable due to privacy or ethical restrictions.

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
