# Peer review of "Dietary and Lifestyle Interventions to Mitigate Oxidative Stress in Male and Female Fertility: Practical Insights for Infertility Management—A Narrative Review"

_metabolites, 2025, doi:10.3390/metabo15060379_

Round 1

Reviewer 1 Report

Comments and Suggestions for Authors

The authors analyzed Dietary and Lifestyle Interventions to Mitigate Oxidative Stress in Male and Female Fertility.

The manuscript is interesting, well written and balanced in its segments.

Here are some comments to improve it before publication :

  • Add the type of study in the title
  • You can add in the title the main aim of the manuscript : practical insights to develop strategies to target infertility
  • Both tables and figures are well made ; you could add one in the first chapters to explain pathophysiology of OS
  • Explain in table 1 : micronutrients , low glycemic diet
  • Add guidelines for regular exercice : duration and types
  • Weight management : target which weight / waist circumference
  • Also alcoohol : types and quantity
  • In stress reduction , authors could add works on oxydative stress , and roles of mediterranean diet to reduce it
  • Table 4 : details doses of each components

Reviewer 2 Report

Comments and Suggestions for Authors

Review x

Dear authors, this is a major issue of great interest. I have some comments:

I disagree with your affirmation in the abstract:  Infertility in both men and women is primarily caused 19 by oxidative stress (OS).

However, OS is a major cause of infertility, in my opinion.

Introduction is well presented.

It is unclear what kind of review did you conduct. You said in the abstract that it is a comprehensive review, how did you make the search, what kind of tools did you use? How many manuscripts have you find and how did you choose the relevant articles?

In Section 2 , related to the implication of ROS in infertility pathogenesis, I suggest you to include a table with the role of ROS, the specific role in infertility, how they affect the quality of oocyte and sperm.

Subsection 4.2. Weight Management I think is more suitable to section  3. Dietary Interventions

In Figure 3 the authors recommend biomarkers. Are they cost/effective?

When introducing the key words in Google Schoolar only 1680 titles are generated, I think more references are needed to be included.

Otherwise, the manuscript is of good value. Congratulations!

Reviewer 3 Report

Comments and Suggestions for Authors

In the present paper, Efthalia Moustakli and coworkers conducted a comprehensive review of clinical, translational, and molecular studies exploring the mechanisms by which oxidative stress affects fertility, as well as the efficacy of nutritional and behavioral strategies. The authors concluded that fertility in both sexes can be improved, and oxidative stress significantly reduced, through a multimodal approach that incorporates dietary and lifestyle changes. Of course, future studies should concentrate on standardized procedures and long-term results. Overall, I think that the manuscript is timely, well-written (within the scope of this journal) and well-structured; then, I would like to congratulate the authors. So far, I raise a series of points to address carefully to improve, in my humble opinion, the overall quality of paper.

1) Please clarify the type of review (comprehensive) in the title of revised paper.

2) Recent evidence suggested that plant-based diets are particularly abundant in nutraceuticals and/or functional foods with antioxidant and anti-inflammatory activity, that could have a crucial, multifaceted, positive role in fighting noncommunicable diseases, also improving the reproductive health. Please deeply discuss this very intriguing topic of current research (for your convenience, you could consider: (BMC Pregnancy Childbirth, 2022, 22, 787; Nutrients, 2022, 14, 1550).

3) Functional foods, nutraceuticals, or dietary supplements, in the context of healthy diet and an adequate physical activity, should address the impact on the microbioma of food and all potential interactions with a preventive and/or therapeutic intervention. Please discuss this intriguing topic in the revised version of paper (see for your convenience: Br. J. Pharmacol. 2020, 177, 1351-1362; Nat Rev Gastroenterol Hepatol 2022, 19, 565-584).

4) NLRP3 inflammasome is one of the major mediators of inflammatory responses, and its activation is closely linked to oxidative stress (Signal Transduct. Target. Ther. 2024, 9, 10) Specifically, in the ovarian milieu, oxidative stress and NLRP3 inflammasome activation interact intricately, and their combined effects on reproductive outcomes are significant. Consequently, NLRP3 inflammasome can be considered an interesting target for innovative bioactive compounds in different organs. Please add a careful comment in the discussion section of the paper, and please kindly insert appropriate references in the revised manuscript.

Round 2

Reviewer 2 Report

Comments and Suggestions for Authors

Dear authors, your manuscript looks better.